# The Anti-Tubercular Aminolipopeptide Trichoderin A Displays Selective Toxicity against Human Pancreatic Ductal Adenocarcinoma Cells Cultured under Glucose Starvation

**DOI:** 10.3390/pharmaceutics15010287

**Published:** 2023-01-14

**Authors:** Johanes K. Kasim, Jiwon Hong, Anthony J. R. Hickey, Anthony R. J. Phillips, John A. Windsor, Paul W. R. Harris, Margaret A. Brimble, Iman Kavianinia

**Affiliations:** 1School of Biological Sciences, The University of Auckland, 3A Symonds Street, Auckland 1010, New Zealand; 2Maurice Wilkins Centre for Molecular Biodiscovery, The University of Auckland, 3A Symonds Street, Auckland 1010, New Zealand; 3Surgical and Translational Research Centre, The University of Auckland, 22-30 Park Avenue, Auckland 1023, New Zealand; 4Department of Surgery, The University of Auckland, 22-30 Park Avenue, Auckland 1023, New Zealand; 5School of Chemical Sciences, The University of Auckland, 23 Symonds Street, Auckland 1010, New Zealand

**Keywords:** aminolipopeptide, trichoderin A, anticancer peptide, pancreatic ductal adenocarcinoma, glucose dependency, cancer metabolism

## Abstract

Pancreatic ductal adenocarcinoma remains a highly debilitating condition with no effective disease-modifying interventions. In our search for natural products with promising anticancer activity, we identified the aminolipopeptide trichoderin A as a potential candidate. While it was initially isolated as an antitubercular peptide, we provide evidence that it is also selectively toxic against BxPC-3 and PANC-1 human pancreatic ductal adenocarcinoma cells cultured under glucose deprivation. This has critical implications for the pancreatic ductal adenocarcinoma, which is characterized by nutrient deprivation due to its hypovascularized network. We have also successfully simplified the trichoderin A peptide backbone, allowing greater accessibility to the peptide for further biological testing. In addition, we also conducted a preliminary investigation into the role of peptide lipidation at the *N*-terminus. This showed that analogues with longer fatty acyl chains exhibited superior cytotoxicity than those with shorter acyl chains. Further structural optimization of trichoderin A is anticipated to improve its biological activity, whilst ongoing mechanistic studies to elucidate its intracellular mechanism of action are conducted in parallel.

## 1. Introduction

Pancreatic cancer is a scourge of modern medicine, manifesting predominantly as pancreatic ductal adenocarcinoma (PDAC) in the clinic [1]. Despite a continual rise in incidence per annum, PDAC grimly remains without any disease-modifying interventions. This is attributed in part to its late diagnosis, such that most patients present with advanced-stage disease, rendering them ineligible for surgery as the primary intervention [2]. The underlying biology of PDAC is further complicated by a desmoplastic stromal compartment, which erects a physical barrier to drug delivery and an immunologically ‘cold’ tumor microenvironment (TME) [3]. However, its dichotomous nature also lends to the inefficient vascularization of the tumor bed. Consequently, cells proximal to the center of the tumour mass are deprived of both nutrient and oxygen supply (Figure 1). The effective management of PDAC therefore demands a combination of pharmaceutics that can address this inherent molecular heterogeneity [4].

Over the last decade, research in our group has focused on the synthesis and biological evaluation of aminolipopeptides as new generation anticancer therapeutics [7]. They are structurally defined by the α,α-dialkylated amino acid 2-aminoisobutyric acid (Aib) and an acylated *N*-terminus. A subset of this family, containing the non-natural building block 2-amino-6-hydroxy-4-methyl-8-oxodecanoic acid (AHMOD), has provided two promising leads in leucinostatin A (**1**) and culicinin D (**2**). Leucinostatin A (**1**) was shown to be cytotoxic against DU-145 human prostate cancer cells in a coculture model with prostate stromal cells (PrSC). It was proposed that this peptide inhibited the transcription of insulin-like growth factor 1 (IGF-1) in PrSC, thereby inhibiting the proliferation of DU-145 [8]. Similarly, culicinin D (**2**) was selectively toxic against MDA-MB-468 human breast cancer cells, via the inhibition of the mTOR signaling pathway [9].

Another member of this family is the octameric peptide trichoderin A (**3**), first isolated by Kobayashi’s group from the marine sponge-derived *Trichoderma* sp. fungus, strain 05FI48 [10]. Follow-up mechanistic studies correlated its antitubercular activity by inhibition of ATP synthesis [11]. However, similarities in structure to leucinostatin A (**1**) and culicinin D (**2**) suggest its potential as an anticancer agent (Figure 2). We therefore sought to examine this hypothesis in the context of PDAC, toward the aim of advancing the collective understanding of the disease biology, in addition to evaluating the prospect of aminolipopeptides as oncotherapeutics more broadly. To this end, we report the synthesis of a simplified trichoderin A analogue (**4**), followed by a small library of its *N*-lipidated analogues (**14**)–(**19**). We also provide their in vitro cytotoxicity profile against two representative human PDAC cell lines BxPC-3 and PANC-1 [12].

## 2. Materials and Methods

### 2.1. Materials

All reagents purchased from commercial vendors were used without additional purification. *N*,*N*-diisopropylethylamine (DIPEA; ReagentPlus grade), piperidine, propionic acid, butanoic acid, hexanoic acid, octanoic acid, decanoic acid, dodecanoic acid, formic acid (LC-MS grade), acetic anhydride, *N*,*N*′-diisopropylcarbodiimide (DIC), and dimethyl sulfoxide (DMSO, cell culture grade) were supplied by Sigma-Aldrich (St Louis, MO, USA). *N*,*N*-dimethylformamide (DMF), dichloromethane (CH_2_Cl_2_), and methanol (MeOH) were purchased from ECP Ltd. (Auckland, New Zealand). Milli-Q water (H_2_O) for RP-HPLC and LC-MS was obtained from the Sartorius Arium Pro ultrapure water production system (Gottingen, Germany). *O*-(7-azabenzotriazol-1-yl)-*N*,*N*,*N’*,*N’*-tetramethyluronium hexafluorophosphate (HATU) and ethyl-2-cyano-2-(hydroxyamino)acetate (Oxyma Pure) were purchased from Novabiochem (Darmstadt, Germany). 1-[(1-(Cyano-2-ethoxy-2-oxoethylideneaminooxy)-dimethylaminomorpholinomethylene)] methanaminium hexafluorophosphate (COMU) and 6-chloro-1-hydroxybenzotriazole (6-Cl-HOBt) were sourced from Aapptec (Louisville, KY, USA). HPLC and LC-MS grade acetonitrile (CH_3_CN) were purchased from Merck (Darmstadt, Germany). Trifluoroacetic acid (TFA) was purchased from Oakwood Chemicals (Estill, SC, USA). 2-Chlorotrityl chloride (2-CTC) polystyrene resin, Fmoc-α-aminoisobutyric acid (Aib), Fmoc-Ile-OH, Fmoc-Val-OH, and Fmoc-Pro-OH were purchased from CS Bio (Shanghai, China). Boc-Pro-OH was purchased from PolyPeptide Group (Zug, Switzerland). Fmoc-Cha-OH, and 1,1,1,3,3,3-hexafluoro-2-propanol (HFIP) were purchased from AK Scientific (Union City, CA, USA). Leucinostatin A was purchased from Santa Cruz Biotechnology, Inc. (Dallas, TX, USA). Trichoderin A, (2*R*)-methyldecanoic acid (MDA), (2*S*,4*S*,6*R*)-2-amino-6-hydroxy-4-methyl-8-oxodecanoic acid (AHMOD) and (*S*)-2-(2-aminopropyl(methyl)amino)ethanol (AMAE) were synthesized in house as previously reported [13]. Nunc MicroWell 96-Well Black Optical-Bottom Plates with Polymer Base, PrestoBlue™ Cell Viability Reagent, CyQUANT™ Direct Cell Proliferation Assay, Roswell Park Memorial Institute (RPMI) 1640 medium (ATCC Modification, #A1049101; 25 mM glucose), RPMI 1640 medium (#11879020; no glucose), Dulbecco’s Eagle Modified Medium (DMEM; high glucose, pyruvate, # 11995065; 25 mM glucose), DMEM (#11966025; no glucose), sodium pyruvate (100 mM), d-glucose solution (200 g/L), antibiotic–antimycotic (A/A), TrypLE™ Express Enzyme (1×) (phenol red, #12605010), and phosphate-buffered saline (PBS, pH 7.4) were purchased from Thermo Fisher Scientific (Waltham, MA, USA). Fetal bovine serum (FBS, heat-inactivated) was purchased from Moregate Biotech (Hamilton, New Zealand).

### 2.2. General Procedure for Peptide Synthesis

The peptides were assembled manually by 9-fluorenylmethoxycarbonyl/*tert*-butyl solid phase peptide synthesis (Fmoc/*t*Bu SPPS) in a fritted glass reaction vessel.

#### 2.2.1. (Method 1) Attachment of Fmoc-Aib-OH on Resin

A solution of Fmoc-Aib-OH (5 eq., 162.7 mg, 0.5 mmol) and DIPEA (10 eq., 172 µL, 1 mmol) in CH_2_Cl_2_ (3 mL) was added to 2-CTC resin (112.4 mg, 0.1 mmol; loading: 0.89 mmol/g) that has been preswollen in CH_2_Cl_2_ (3 mL) for 15 min. The reaction mixture was agitated for 6 h at room temperature, filtered, and repeated with fresh reagents for a further 6 h. Excess solvent was then filtered, and the resin bed treated with a mixture of CH_2_Cl_2_/MeOH/DIPEA (8:1.5:0.5, *v*/*v*/*v*, 3 mL) for 15 min at room temperature to cap unreacted chlorides on the resin.

#### 2.2.2. (Method 2) Spectrophotometric Quantitation of First Residue Loading on Resin

Loading of the first residue on resin was quantified on the Shimadzu UV-1280 UV-VIS spectrophotometer (Kyoto, Japan). Resin that has been dried under vacuum was weighed into a Starna Scientific 10-mm matched silica UV spectrophotometric cuvette (Ilford, UK). 20% piperidine/DMF (*v*/*v*, 3 mL) was dispensed into the cuvette, briefly agitated with a pipettor to achieve a uniform suspension, then left for 5 min at room temperature. A reference cuvette, containing only 20% piperidine/DMF (*v*/*v*, 3 mL), was used to zero the spectrophotometer at the wavelength of 290 nm. The absorbance of the resin-containing cuvette was recorded as an average of three independent measurements, and the loading of the first residue calculated by the following Equation (1):(1)$$\mathrm{Loading}=\frac{\mathrm{Absorbance}}{\left(\mathrm{Resin}\;\mathrm{mass}\times 1.75\right)}$$

Equation (1) estimation of first residue attachment on resin. Loading is measured in mmol/g.

#### 2.2.3. (Method 3) Removal of Fmoc Protecting Group

A solution of 20% piperidine/DMF (*v*/*v*, 3 mL) was added to the peptidyl resin to remove the Fmoc protecting group. The reaction mixture was agitated for 5 min at room temperature, filtered, and the reaction repeated with fresh reagents for a further 10 min. Excess solvent was then filtered, and the resin bed washed with DMF (3 × 3 mL).

#### 2.2.4. (Method 4) Procedure for the Difficult Sequential Aib-Aib Coupling on Resin

A solution of Fmoc-Aib-OH (5 eq., 162.7 mg, 0.5 mmol), COMU (5 eq., 214 mg, 0.5 mmol), Oxyma (5 eq., 71 mg, 0.5 mmol) and DIPEA (10 eq., 172 µL, 1 mmol) in DMF (3 mL) was added to the peptidyl resin (0.1 mmol). The reaction mixture was agitated for 2 h at room temperature, filtered, and the reaction repeated with fresh reagents for a further 2 h. Excess solvent was then filtered and the resin bed washed with DMF (3 × 3 mL).

#### 2.2.5. (Method 5) Coupling of Fmoc-Protected Amino Acid

A solution of the appropriate Fmoc-Aa-OH (5 eq., 0.5 mmol), HATU (4.9 eq., 186 mg, 0.49 mmol) and DIPEA (10 eq., 172 µL, 1 mmol) in DMF (3 mL) was added to the peptidyl resin (0.1 mmol). The reaction mixture was agitated for 1 h at room temperature, filtered, and the reaction repeated with fresh reagents for a further 1 h. Excess solvent was then filtered, and the resin bed was washed with DMF (3 × 3 mL).

#### 2.2.6. (Method 6) Attachment of *N*-Terminal Fatty Acid

A solution of the appropriate fatty acid (5 eq., 0.5 mmol), COMU (5 eq., 214 mg, 0.5 mmol), Oxyma (5 eq., 71 mg, 0.5 mmol) and DIPEA (10 eq., 172 µL, 1 mmol) in DMF (3 mL) was added to the peptidyl resin (0.1 mmol). The reaction mixture was agitated for 3 h at room temperature. Excess solvent was then filtered, the resin bed washed with DMF (3 × 3 mL) and CH_2_Cl_2_ (3 × 3 mL), then dried under vacuum.

#### 2.2.7. (Method 7) HFIP-Mediated Resin Cleavage

The resin bed was treated with 20% HFIP/CH_2_Cl_2_ (*v*/*v*, 5 mL) to release the completed peptide chain. The reaction mixture was agitated for 30 min at room temperature, filtered, and the reaction repeated with fresh reagents for a further 30 min. The collected filtrate was partially concentrated under a gentle stream of nitrogen, reconstituted in CH_3_CN/H_2_O (*v*/*v*, 10 mL) and lyophilized.

#### 2.2.8. (Method 8) Late-Stage Solution Phase *C*-Terminal Coupling of AMAE

The *C*-terminal AMAE residue was conjugated to the peptide as a TFA salt. A solution of AMAE.2TFA (3 eq.), DIC (6 eq.), 6-Cl-HOBt (6 eq.), and DIPEA (6 eq.) in DMF was added to the purified peptide from Method 7. The reaction mixture was agitated for 12 h at room temperature and completion of the reaction was monitored by analytical RP-HPLC and ESI-MS. The peptide mixture was then purified batchwise by semipreparative RP-HPLC.

#### 2.2.9. LC-MS Analysis of Purified Peptides

LC-MS spectra were acquired on an Agilent Technologies (Santa Clara, CA, USA) 1260 Infinity LC equipped with an Agilent Technologies 6120 Quadrupole mass spectrometer, using an Agilent C3 analytical column (150 mm × 3.0 mm, 3.5 µm) at a flow rate of 0.3 mL min^−1^. A linear gradient of 5% to 95% B was used over 30 min (ca. 3% B min^−1^), where solvent A was 0.1% formic acid in H_2_O and B was 0.1% formic acid in CH_3_CN.

#### 2.2.10. Semipreparative RP-HPLC Purification of Crude Peptides

Purification of the crude peptides by RP-HPLC was performed on a Thermo Scientific (Waltham, MA) Dionex Ultimate 3000 HPLC equipped with a four-channel UV Detector at 210, 225, 254, and 280 nm using a Waters (Milford, MA, USA) XTerra^®^ MS C18 semipreparative column (10 × 250 mm, 5 µm) at a flow rate of 4 mL min^−1^. A linear gradient of 20% to 95% B was used over 75 min (ca. 1% B min^−1^), where solvent A was 0.1% TFA in H_2_O and solvent B was 0.1% TFA in CH_3_CN.

#### 2.2.11. Analytical RP-HPLC of Purified Peptides

Analysis of the purified peptide fractions by RP-HPLC was performed on a Thermo Scientific Dionex Ultimate 3000 HPLC equipped with a four-channel UV Detector at 210, 225, 254, and 280 nm using a Waters XTerra^®^ MS C18 analytical column (4.6 × 150 mm, 5 µm) at a flow rate of 1 mL min^−1^. A linear gradient of 5% to 95% B was used over 30 min (ca. 3% B min^−1^), where solvent A was 0.1% TFA in H_2_O and solvent B was 0.1% TFA in CH_3_CN. Fractions with the correct *m*/*z* as determined by ESI-MS were pooled and lyophilized.

### 2.3. Cell Culture

BxPC-3 (ATCC^®^ CRL-1687™) and PANC-1 (ATCC® CRL-1469™) human PDAC cell lines were purchased from the American Type Culture Collection (Rockville, MD, USA). BxPC-3 was cultured in RPMI-1640 (ATCC Modification) supplemented with 10% FBS (*v*/*v*) and 1× A/A. PANC-1 was cultured in DMEM (high glucose, pyruvate) supplemented with 10% FBS (*v*/*v*) and 1× A/A. Both cell lines were maintained in culture at 37 °C, 5% CO_2_, and harvested at ~90% confluency using TrypLE Express and subcultured into a new flask at a splitting ratio of 1:6 (BxPC-3) or 1:4 (PANC-1). Culture medium was refreshed 2 times per week, and cell growth monitored daily under an inverted microscope.

### 2.4. Initial Cytotoxicity Assay: BxPC-3

The sensitivity of BxPC-3 cells to the native trichoderin A (**3**) and its simplified analogue (**4**) was examined under aerobic conditions in vitro by means of the orthogonal, fluorometric PrestoBlue™ and CyQUANT™ Direct Confirmation Assay [14]. BxPC-3 cells were seeded at a density of 7500 cells/well in Thermo Fisher Scientific (Waltham, MA, USA) Nunc MicroWell 96-Well Black Optical-Bottom Plates with Polymer Base (total volume: 100 µL). Cells were first incubated overnight at 37 °C, 5% CO_2_ to allow for adherence, and then exposed to both peptides (**3**) and (**4**) in duplicate at a concentration range of 50.8 pM to 1 µM for 72 h (threefold serial dilution in RPMI-1640 (ATCC modification, 25 mM glucose; ten steps). At the end of this incubation step, 11 µL of PrestoBlue™ Cell Viability Reagent was added into each well, and the plates incubated for 10 min. The endpoint fluorescence was read from the bottom of the plate at an excitation/emission (Ex/Em) wavelength of 560/590 nm on a SpectraMax iD3 Multi-Mode Microplate Reader (Molecular Devices, San Jose, CA, USA). The CyQUANT™ Direct Cell Proliferation Assay was subsequently performed by mixing the CyQUANT™ Direct nucleic acid stain, CyQUANT™ Direct background suppressor I, and PBS (pH 7.4) as per the vendor instructions [15], then adding 100 µL of this mixture into each well. The plate was returned into the incubator for a further 1 h, and the endpoint fluorescence bottom-read at the Ex/Em wavelength of 485/538 nm on the SpectraMax iD3 plate reader. Blank wells (medium only) and two negative controls—(1) cells + medium, (2) cells + medium + 0.3% *v*/*v* DMSO (equivalent to 1 µM peptide)—were set up in the plate for each experiment.

### 2.5. Comparative Cytotoxicity Assay to Evaluate the Effect of Glucose on Cellular Proliferation: BxPC-3 and PANC-1

The sensitivity of BxPC-3 and PANC-1 cells to the native trichoderin A (**3**) and its analogues (**4**), (**12**)–(**19**) under low and normal glucose conditions was examined in parallel on the same plate using the PrestoBlue™ and CyQUANT™ Direct Confirmation Assay. Cells were seeded at a density of 25,000 cells/well (BxPC-3) and 30,000 cells/well (PANC-1) in Nunc MicroWell 96-Well Black Optical-Bottom Plates with Polymer Base (total volume: 100 µL). Cells were first incubated overnight at 37 °C, 5% CO_2_ to permit adherence, after which the medium in each well was aspirated, briefly washed once with 100 µL/well of PBS (pH 7.4) and replaced with 100 µL of either RPMI 1640 (BxPC-3, #11879020) or DMEM (PANC-1, #11966025) supplemented with or without 5 mM glucose to create the ‘normal glucose’ and ‘low glucose’ media, respectively. This 5 mM glucose concentration was selected to reflect the physiological concentration of glucose in plasma [16]. Each well was then exposed to the peptides in duplicate at a concentration range of 50.8 pM to 1 µM (BxPC-3) or 457 pM to 9 µM (PANC-1) for 24 h (threefold serial dilution in the corresponding ‘normal glucose’ or ‘low glucose’ media, ten steps). Endpoint fluorescence was quantified as described above.

### 2.6. Data Analysis

Blank correction (wells containing medium only) was first performed on the raw fluorescence–concentration data from the plate reader. Cell viability was then calculated as a percentage by normalizing to the fluorescence of the wells containing the lowest drug concentration (50.8 pM for BxPC-3, 457 pM for PANC-1). The resulting data from three independent experiments (each in technical duplicate) was then plotted on GraphPad Prism 9 and analyzed using the built-in ‘[Inhibitor] vs. response—Variable slope (four parameters)’ equation. Cytotoxicity of each peptide reported in terms of:IC_50_, corresponding to the drug concentration that gave half-maximal response (i.e., 50% viability)AUC, corresponding to the calculated area under the curve with baseline set at y = 0 and the full concentration range used for both cell lines (i.e., 50.8 pM–9 µM).E_max_, corresponding to the maximal response achieved by the drug at the highest concentration used in the assay (i.e., 1 µM for BxPC-3, 9 µM for PANC-1).

## 3. Results

### 3.1. Chemical Synthesis of the Simplified Trichoderin A Analogue (4)

The first total synthesis of trichoderin A (**3**) was reported by our group [13]. Within this work, the absolute stereochemistry of the C-6 atom of AHMOD was also revised to (*R*) from the initially reported (*S*) [10]. In a similar vein to culicinin D (**2**), the routine synthesis of trichoderin A (**3**) is also bottlenecked by the AHMOD building block, which preparation entails multiple complex reaction steps, and only afforded the desired product in abysmally low yields [17,18]. Thus, it was deemed necessary to identify a suitable replacement to AHMOD that would afford an equipotent analogue and concomitantly expedite the routine synthesis of the peptide for downstream biological studies. Fortuitously, in our preceding work with culicinin D (**2**), we have demonstrated that the replacement of AHMOD with l-cyclohexylalanine (Cha) produced such an analogue, as judged by its IC_50_ value from the sulphorhodamine B (SRB) assay against a panel of four human cancer cell lines: MDA-MB-468, SK-BR-3, T47D (breast), and NCI-H460 (lung) [19]. On this basis, we envisaged that this substitution would also be well-tolerated by trichoderin A (**3**), by virtue of its structural similarities to culicinin D (**2**).

To this end, we employed a hybrid solid-/solution-phase synthesis strategy that had proved successful for the preparation of the native trichoderin A (**3**) [13]. The synthesis commenced with the loading of Fmoc-Aib-OH on 2-chlorotrityl chloride (2-CTC) resin (loading: 0.89 mmol/g) (**5**) by Method 1 to give (**6**). Loading was quantified at 0.43 mmol/g by Method 2 (Figure 1).

Following removal of the Fmoc temporary protecting group by Method 3, Fmoc-Aib-OH was coupled using Method 4 to generate the Aib-Aib dipeptide unit (**7**). The peptide chain was subsequently elongated by iterative cycles of Fmoc deprotection (Method 3) and Fmoc-Aa-OH acylation (Method 5) to give (**8**). Method 4 was again revisited for the formation of the second Aib-Aib dipeptide unit at positions 3 and 4 (**9**). The peptide chain (**10**) was completed after coupling of Fmoc-Pro-OH (Method 5) and MDA (Method 6). Cleavage of the peptide from the solid support was undertaken using Method 7, and the crude product (**11**) lyophilized prior to purification by semipreparative RP-HPLC, affording the desired precursor as a white amorphous solid (25.6 mg, 25.9% yield). The *C*-terminal aminoalcohol AMAE was lastly conjugated to the purified precursor (25.6 mg, 0.03 mmol) as AMAE.2TFA salt using Method 8, affording the simplified trichoderin A analogue (**4**) (17.3 mg, 15.7% overall yield based on 0.l mmol resin loading, >99% purity after semipreparative RP-HPLC purification, Figure 3).

### 3.2. In Vitro Antiproliferative Testing of the Simplified Trichoderin A Analogue (4)

The in vitro cytotoxicity of trichoderin A (**3**), prepared as detailed previously [13], and its simplified analogue (**4**) were subsequently tested against the human PDAC cell line BxPC-3 using the orthogonal, fluorometric PrestoBlue™ and CyQUANT™ Direct Confirmation Assay [14]. The PrestoBlue™ reagent utilizes a cell-permeant resazurin-based dye, which is reduced by reductase enzymes present in viable cells, and therefore provides a proxy of their metabolic health [20]. On the other hand, while the CyQUANT™ Direct nucleic acid stain would bind to the DNA of all cells, the accompanying background suppressor blocks the staining of dead or membrane-permeabilized cells. Thus, only the DNA of viable cells will be stained [15]. The orthogonality of this protocol is based on the nonoverlapping excitation/emission wavelength range of the two dyes, allowing for both assays to be performed sequentially on the same plate. To our surprise, both viability assays indicated that both trichoderin A (**3**) and its simplified analogue (**4**) only partially inhibited the proliferation of BxPC-3 cells after a 3-day incubation period, up to the highest concentration level tested of 1 µM (Figure 4).

Intriguingly, there exists a precedence for this observation: the recent discovery of the leucinostatin Y by Kawada’s group was also reported alongside a comparative 2,5-diphenyl-2*H*-tetrazolium bromide (MTT) assay against the parent peptide leucinostatin A (**1**) on the human PDAC cell lines PANC-1, BxPC-3, PSN-1, and PK-8 [21]. Their results indicated that (**1**) is selectively cytotoxic against all four cell lines, but only when they are cultured under minimal glucose in the growth medium. On this basis, we opted to re-evaluate the cytotoxicity of both trichoderins (**3**) and (**4**) against BxPC-3 and also PANC-1 under both ‘low glucose’ and ‘normal glucose’ conditions. Pleasingly, we were able to demonstrate evidence of selective toxicity under the ‘low glucose’ condition in both cell lines (Figure 5 and Table 1), and that the trichoderin A analogue (**4**) was similarly potent to the parent peptide (**3**), thereby validating our rationale for the replacement of AHMOD with Cha.

### 3.3. The Effect of N-Lipidation on the Bioactivity of the Simplified Trichoderin A Analogue (4)

Having established a more synthetically tractable analogue of trichoderin A (**3**), we subsequently decided to explore the role of the *N*-terminus fatty acyl chain on its bioactivity. This study was initiated through the synthesis of trichoderin A analogues (**12**)–(**19**), varying in lipid chain length from C0 (**12**) to C12 (**19**) (Figure 6), following the protocol described in the preceding Figure 1 (see also Appendix A).

In addition to this, we also sought to use the decanoyl analogue (**18**) to probe the functional significance of the methyl group on the C-2 atom of the 2-MDA moiety of analogue (**4**). The synthesized analogues (**12**)–(**19**) were also subjected to an in vitro cytotoxicity screen against BxPC-3 and PANC-1 cells under ‘low glucose’ and ‘normal glucose’ conditions in parallel. Resulting dose–response curves and relevant metrics obtained from culturing both cells under ‘low glucose’ are shown in Figure 7 and Table 1 (see also Appendix A).

The functional significance of lipidation at the *N*-terminus of trichoderin A (**3**) was emphasized by the inactivity of analogue (**12**), which lacked a fatty acyl moiety, across both BxPC-3 and PANC-1 cells, irrespective of the culture condition. Intriguingly, the presence of an acetyl group on (**13**) was already sufficient to induce cytotoxic effects against PANC-1, but not in BxPC-3. In general, we noted a gradual improvement in compound potency with increasing hydrocarbon length of the fatty acyl moiety, although the octanoyl analogue (**17**) was curiously less potent than the preceding hexanoyl analogue (**16**) against both cell lines. This trend is consistent across the three response metrics used to quantify the cytotoxicity profile of each drug, namely IC_50_, AUC, and E_max_. We also frequently noted higher cell viabilities from the CyQUANT™ Direct assay than the PrestoBlue™ assay, which could suggest that some viable cells with intact DNA may be metabolically impaired, and thus considered nonviable by the PrestoBlue™ assay. The decanoyl analogue (**18**) was also satisfyingly found to be more potent than analogues (**3**) and (**4**), suggesting that the removal of the C-2 methyl functionality was also well tolerated with regards to the bioactivity of trichoderin A (**3**). Collectively, the dodecanoyl analogue (**19**) is the most potent analogue in this tested series, and thus presents as a new template for further structural optimizations toward a more potent trichoderin A (**3**) analogue.

## 4. Discussion

Cancer cells preferentially generate ATP via the fermentation of glucose to lactate, compensating for the relative inefficiency of this pathway to the mitochondrial oxidative phosphorylation (OXPHOS) pathway by increasing both glucose uptake and lactate secretion [22]. This observation, first reported by the German physiologist Otto Warburg in isolated tumor tissues, formed the basis for the ‘Warburg effect’ [23]. While initially thought to be due to mitochondrial dysfunction, certain cancer subtypes, including PDAC, have been established to contradict this notion by having intact mitochondria, and thus remain capable of producing ATP via OXPHOS (Figure 8) [24].

Our selection of BxPC-3 and PANC-1 in this study was based on their differential metabolic phenotypes, as established by Piccoli’s group [25]. Unlike this study, however, they opted to induce this condition via substitution with galactose, which is metabolized at a slower rate to glucose. Consequently, both cells underwent a forced shift to aerobic respiration by OXPHOS. The dependency of BxPC-3 on glycolysis meant that its proliferation was significantly halted when glucose became a limiting factor. In contrast, the growth of the OXPHOS-competent PANC-1 was only moderately stunted, perhaps reflecting a greater degree of flexibility to seek alternative fuel sources, such as glutamine [26], to maintain the biochemical flux of the tricarboxylic acid (TCA) cycle. Findings from this work was also recently reinforced by Carrier’s group, who carried out a comparative bioenergetics analysis of six human PDAC cell lines, including BxPC-3 and PANC-1. Their results also classified both cell lines under the ‘glycolytic’ and ‘OXPHOS’ groups, respectively [27].

Our results indicate that trichoderin A (**3**) and its *N*-lipidated analogues (**12**)-(**19**) are universally more potent against BxPC-3 than PANC-1 cells. Given the glycolytic phenotype of BxPC-3 [25,27], the depletion of glucose in the culture medium, which is a requirement for the peptides to be cytotoxic, evidently affected its proliferation at a greater extent relative to PANC-1, which would likely favor the forced shift to OXPHOS-mediated respiration. The dependency of BxPC-3 on glucose is further emphasized by its inability to adapt to this mode of respiration, resulting in a G2/M cell cycle shift and consequently death by apoptosis [25]. Another potential angle, while beyond the scope of the present study, is the differential mutation status of the Kirsten ras oncogene homologue (*KRAS*) between the two cell lines: BxPC-3 is wild-type (WT) for this gene, whereas PANC-1 carries a G12D mutation, which is the most common subtype in PDAC and leads to the worst prognosis [28]. While it can be rightly argued that PANC-1 is a more representative cell line to evaluate prospective agents against PDAC, the disease unsurprisingly has a *KRAS* WT subtype, which would likely require differential management [29]. Evidently, a ‘one-size-fits-all’ approach is unlikely to be appropriate for the effective management of PDAC, which future treatment options lie on a therapeutic regimen that is personalized for each patient, ensuring maximal improvements in their quality of life [30].

## 5. Conclusions

We have reported herein the successful preparation of a simplified analogue of the antitubercular peptide trichoderin A (**3**), wherein the synthetically challenging AHMOD building block has been substituted by the commercially available Cha to afford analogue (**4**). Furthermore, we have also provided, to the best of our knowledge, the first report of its selective antiproliferative activity against two human PDAC cell lines (BxPC-3 and PANC-1) cultured under minimal glucose. While these results are undoubtedly intriguing and lay the foundation for further studies into this unique aminolipopeptide family, it must be duly noted that cell lines grown in monolayer are not a faithful recapitulation of the complex architecture of the PDAC tumor, as briefed in the Introduction. Therefore, the reproduction of these results in higher order in vitro models such as three-dimensional PDAC organoids [31] is anticipated to advance our understanding of the pharmacology of these aminolipopeptides. The conditional cytotoxicity of trichoderin A (**3**) also raises some crucial questions on the translatability of these findings in vivo, wherein the subpopulation of cells that are glucose deprived are often reasonably distal from the vasculature, and importantly also hypoxic. Consequently, there likely will be a prerequisite to first arrest glucose uptake/metabolism locally in the tumor prior to administration of the peptide drugs. We envisage that the use of pan-glucose transporter (GLUT) inhibitors such as Glutor [32] in combination with the dodecanoyl analogue of trichoderin A (**19**) may assist in this cause.

Future work on this aminolipopeptide should also involve the preparation of suitable analogues to explore the potential of other biologically relevant moieties, such as its *C*-terminal aminoalcohol, which we have shown previously in culicinin D (**2**) to be an equally critical contributor to its activity [33]. Lastly, relevant studies toward the elucidation of the intracellular mechanism of action of trichoderin A (**3**) are currently underway, within the context of its selective cytotoxicity under glucose deprivation. Inhibition of the OXPHOS machinery appears to be the central mode of action uniting both leucinostatin A (**1**) and trichoderin A (**3**), and could present as a novel angle by which to target nutrient-deprived cancer cells, which are generally more resistant to chemotherapy.

## Data Availability

All data can be provided by the authors upon request. No publicly accessible archive storage is available.

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
