# Peer review of "The Anti-Tubercular Aminolipopeptide Trichoderin A Displays Selective Toxicity against Human Pancreatic Ductal Adenocarcinoma Cells Cultured under Glucose Starvation"

_pharmaceutics, 2023, doi:10.3390/pharmaceutics15010287_

Round 1
Reviewer 1 Report (Previous Reviewer 2)
The authors have addressed my concerns raised during previous submission. I have no more comments.
Reviewer 2 Report (Previous Reviewer 1)
I think the manuscript could be considered for publication in the present form.
This manuscript is a resubmission of an earlier submission. The following is a list of the peer review reports and author responses from that submission.
Round 1
Reviewer 1 Report
The manuscript entitled "The anti-tubercular aminolipopeptide trichoderin A displays selective toxicity against human pancreatic ductal adenocarcinoma cells cultured under glucose deprivation" describes the synthesis and characterization of the anticancer activity of lipopeptides. I think the theme of the manuscript fits the scope of the journal. Also, the manuscript is well written and the data are well organized and support the hypothesis. Therefore, I would recommend acceptance as the present form.
Author Response
We kindly thank all reviewers for the helpful comments and/or suggestions provided for our manuscript “The anti-tubercular aminolipopeptide trichoderin A displays selective toxicity against human pancreatic ductal adenocarcinoma cells cultured under glucose deprivation”, submitted to MDPI Pharmaceutics journal under the ‘Drug Targeting and Design’ section, special topic ‘Peptide-Based Drugs for Cancer Therapies’.
Herein we address each of the reviewer’s comments in detail:
Reviewer 1:
The manuscript entitled “The anti-tubercular aminolipopeptide trichoderin A displays selective toxicity against human pancreatic ductal adenocarcinoma cells cultured under glucose deprivation” describes the synthesis and characterization of the anticancer activity of lipopeptides. I think the theme of the manuscript fits the scope of the journal. Also, the manuscript is well written and the data are well organized and support the hypothesis. Therefore, I would recommend acceptance as the present form.
We thank the reviewer for their kind comments, and for recommending the acceptance of the manuscript for publication in MDPI Pharmaceutics in its current form.
Reviewer 2 Report
The article by Kasim et al described the synthesis and preliminary in vitro cytotoxicity evaluation of simplified trichoderin A analogue and its N-lipidated derivatives in BxPC-3 line. This study is mostly focused on the synthesis and characterization of the peptidomimetics which has been done meticulously. However, the study is lacking in the areas of biological evaluation and impact in therapeutic perspective as outlined below.
- First of all, more than 90% of PDAC harbors Kras mutation, but the authors used BxPC3 cell line which is Kras wild type. Use of other PDAC cell lines with Kras mutation will increase the impact of this study, especially considering the role of Kras in glucose uptake and metabolism.
- A potential limitation of this strategy is its applicability in vivo. As shown by the authors, these analogues are only effective in glucose-deprived conditions, and these glucose deprived conditions are present mostly at the core of the tumors with inefficient vascularization. Hence, there is a possibility that these analogues won’t be able to reach the core where they should have been the most effective. On the other hand, the periphery of the tumor will have comparatively more vasculature and hence more availability of these analogues, but those cells will not be glucose-deprived, making these analogues ineffective.
- Almost all of the synthesized analogues are significantly less effective than leucinostatin A, which itself did not generate much enthusiasm as cancer therapeutics, thus raising concern about the impact of this study.
Author Response
Reviewer 2:
The article by Kasim et al described the synthesis and preliminary in vitro cytotoxicity evaluation of simplified trichoderin A analogue and its N-lipidated derivatives in BxPC-3 line. This study is mostly focused on the synthesis and characterization of the peptidomimetics which has been done meticulously. However, the study is lacking in the areas of biological evaluation and impact in therapeutic perspective as outline below.
- First of all, more than 90% of PDAC harbors Kras mutation, but the authors used BxPC3 cell line which is Kras wild type. Use of other PDAC cell lines with Kras mutation will increase the impact of this study, especially considering the role of Kras in glucose uptake and metabolism.
We are in full agreement with the points raised by the reviewer. Indeed, the impact of this study will be further strengthened by additional validation work in Kras-mutant PDAC cell lines, such as PANC-1. However, we have recently purchased three PDAC organoids (one Kras wild type and two Kras mutants) as an additional tool for our compound screening programme. We believe that the 3D arrangement of this model, driven by developmental cues from the extracellular matrix scaffold and PDAC-specific growth medium, would provide a more holistic representation of the tumour as it manifests in humans, thus allowing for a more accurate quantification of drug response in vitro prior to commencing in vivo animal studies. To this end, we intend to assess the efficacy of our lead trichoderin A analogue (19) against both Kras wild type and mutant PDAC organoids and compare the results with the corresponding 2D PDAC cell lines.
- A potential limitation of this strategy is its applicability in vivo. As shown by the authors, these analogues are only effective in glucose-deprived conditions, and these glucose deprived conditions are present mostly at the core of the tumors with inefficient vascularization. Hence, there is a possibility that these analogues won’t be able to reach the core where they should have been the most effective. On the other hand, the periphery of the tumor will have comparatively more vasculature and hence more availability of these analogues, but those cells will not be glucose-deprived, making these analogues ineffective.
We are in full agreement with the points raised by the reviewer. The delivery of the lead analogue will no doubt be an exceptional challenge, given the well-documented notoriety of PDAC. While the investigation of such strategies is beyond the scope of the current research, we have been actively assessing potential modalities such as conjugation to a monoclonal antibody to generate an antibody-drug conjugate. We are also pleased to disclose that we are conducting preliminary experiments to support this aim, which we anticipate will form the basis of a follow-up publication.
- Almost all of the synthesized analogues are significantly less effective than leucinostatin A, which itself did not generate much enthusiasm as cancer therapeutics, thus raising concern about the impact of this study.
While the trichoderin A analogues in this study are generally less potent than leucinostatin A, with the lead analogue (19) being approximately four-fold less active, we must direct the reviewer’s attention to the ease of synthesis of these peptides. Leucinostatin A possesses five non-natural building blocks which complicates its routine synthesis: 4-methyl-2-hexenoic acid, 4-methylproline, 2-amino-6-hydroxy-4-methyl-8-oxodecanoic acid (AHMOD), β-hydroxyleucine, and N,N-dimethylpropane-1,2-diamine. In contrast, trichoderin A only has two: AHMOD and 2-(2-aminopropyl(methyl)amino)ethanol (AMAE). We have shown in the present work that substitution of AHMOD with Cha afforded a bioequivalent trichoderin A analogue (4), thus eliminating the need for its preparation and allowing for larger scale synthesis. While the substitution of AHMOD with Cha may also generate a bioequivalent leucinostatin A analogue, further structure-activity relationship (SAR) studies are still required to identify suitable replacements for the other four non-natural building blocks. We are also currently exploring other substitutions on the lead trichoderin A analogue (19) to improve upon its bioactivity, as well as designing suitable biological assays to elucidate its differential mechanism of action based on extracellular glucose availability.
Reviewer 3 Report
Kasim J.K. proposed an anti-tubercular aminolipopeptide, trichoderin A, as anti-pancreatic cancer drug, in particular in glucose deprivation conditions.
The paper is well written and the workflow is clear.
I have only to rise a minor point:
Paragraph 3.1 is a little bit long and this leads to a difficult reading. I suggest to rewritten the paragraph in a more clear way or divided it in two paragraphs: the first one regarding peptide synthesis and the second one the in vitro analysis.
Author Response
Reviewer 3:
The paper is well written and the workflow is clear. I have only to rise a minor point: Paragraph 3.1 is a little bit long and this leads to a difficult reading. I suggest to rewritten the paragraph in a more clear way or divided it in two paragraphs: the first one regarding peptide synthesis and the second one the in vitro analysis.
We thank the reviewer for the suggestion. Having reviewed the manuscript again, we agree that Section 3.1 can be quite a lengthy read. As such, we have opted to split this section into two separate sections, as reflected in the revised manuscript. Therefore, the ‘Results’ section now consists of three subsections:
3.1 Chemical synthesis of the simplified trichoderin A analogue (4)
3.2 In vitro antiproliferative testing of the simplified trichoderin A analogue (4)
3.3 Synthesis and antiproliferative testing of the N-lipidated trichoderin A analogues (12)-(19)
Round 2
Reviewer 2 Report
While the authors addressed some of the concerns raised in prior review, they still didn't include data in at least one Kras mutant pancreatic cancer cell line such as PANC1. It is not that difficult to perform the in vitro experiments in at least one cell line in glucose-depleted and glucose-repleted conditions. This remains a critical omission as outlined in the prior review.
